# Effect of Polycarboxylic Grinding Aid on Cement Chemistry and Properties

**DOI:** 10.3390/polym14183905

**Published:** 2022-09-19

**Authors:** Jianyu Yang, Guanglin Li, Weijun Yang, Junfeng Guan

**Affiliations:** 1College of Civil Engineering, Changsha University of Science & Technology, Changsha 410082, China; 2School of Civil Engineering and Communication, North China University of Water Resources and Electric Power, Zhengzhou 450045, China

**Keywords:** polycarboxylic acid, three polar groups, compounding, grinding aids, cement properties

## Abstract

In view of the disadvantages of polycarboxylic acid grinding aids, such as poor reinforcement effect and cumbersome synthesis process, a new type of polycarboxylic acid grinding aid was prepared to meet the requirements of multifunctional admixture for cement concrete. The polycarboxylate grinding aid (PC) was prepared using acrylic acid, sodium allyl sulfonate, and isoprenol polyoxyethylene ether (TPEG) as raw materials, and ammonium persulfate as initiator in the nitrogen atmosphere. The effect of PC and its compound with triethanolamine (TEA) and triisopropanolamine (TIPA) on cement particle size and strength, and hydration process and structures of hydrated products were investigated. Moreover, the grinding mechanism of grinding aids was also proposed. The results indicate that the PC has good performance in both grinding and high-efficiency water-reducing. The average particle diameter of cement was reduced by 3.65 μm when 0.03 wt% of PC was added as grinding aid. Moreover, a high initial fluidity of the cement paste, 290 mm, could be reached when 0.08 wt% of PC was added. The fluidity loss of cement paste after 30 min and 60 min was 265 mm and 260 mm, respectively. After PC compounding with TEA and TIPA, 4.07 μm and 4.7 μm of the average particle size of the cement can be reduced, respectively. Based on the investigations on the hydration rate of cement hydration, the phases, and the microstructures of the hardened slurry, it could be concluded that grinding aids can change the hydration process of cement and improve the morphologies and structures of hydration products without influence on the type of hydrated products. Note that the compounded grinding aids, such as PC with TEA or PC with TIPA, can more effectively enhance the early and late strength of cement. This shows excellent comprehensive performance. In this study, a new type of polycarboxylic acid grinding aid was prepared to meet the requirements of the versatility of cement concrete additives, and to simplify the synthesis process, reduce production costs, improve the grinding effect, and improve the performance of cement concrete.

## 1. Introduction

Currently, an environment-friendly green development system with a saved resource is increasingly attracting attention, as it is the dominant trend for industrial development. Cement, as a very basic raw material for infrastructure, consumes high energy during its production process [1,2,3]. Note that a large amount of energy was consumed in order to grind cement into the desired particles [4,5]. Chemical admixtures, widely used in cement and concrete, have changed the grinding kinetics and hydration kinetics, and improved the microstructural evolution and performance of cement, leading to reduced energy consumption and establishment of the organic cement chemistry [6].

At present, the most widely used grinding aids are generally composed of a mixture of one or several small molecule compounds, such as alcamines, organic compounds, inorganic salts, and industrial by-products [7]. Investigations revealed that compounds containing more than two groups of polar present better grinding performance than those made from single polar. Moreover, grinding performance is also better with the increased polar groups and the larger structure of these non-polar groups [8]. The combination of TEA and TIPA can effectively improve the friability of clinker and significantly increase the specific surface area of cement products as well as the grindability index [9,10], but it increased the price of the product. Additionally, TEA and TIPA, in general, were used together with inorganic salts as combinations, and a large quantity of inorganic salts in cement can obviously have a negative influence on the durability and soundness of concrete [11]. Therefore, to develop grinding aids with moderate cost and good stability is currently of significance.

It is well known that a variety of chemical additives with some dedicated functions were used in cement and concrete to enhance their performance. For example, air-entraining agent for control of the air content in concrete can improve the fluidity of fresh concrete, and enhance the frost resistance, and durability [12]. For the production of cement concrete, it is essential to use chemical additives, especially shrinkage mitigators and superplasticizers that allow the reduction of the w/c factor without loss of workability [13]. Water reducers can be used to increase the working performance of cement concrete and reduce the amount of water used in the mix etc. [14] Additionally, those additives will also result in changes in other properties, for instance, rheology, setting time, strength development, and cracking behavior, for concrete [15]. Therefore, an investigation concerning the effect of admixtures on the performances of cement will play a very key guiding role in the practical application.

Polycarboxylic acid superplasticizer (PCE) is the third generation of high-performance superplasticizer developed after the ordinary superplasticizer represented by wood calcium and the high-efficiency superplasticizer represented by naphthalene [16]. Polycarboxylic acid is an important surfactant, due to its unique comb-type molecular structure composed of polar functional group branches and non-polar backbones. As a grinding acid, Polycarboxylic with strong tunable molecular structure has been continuously developed to enable low shrinkage and simplified production process of concrete without pollution. Clay often has severe detrimental impacts on cement-based materials. Therefore, it is necessary to investigate the mechanism causing the deterioration to improve the service life of cement-based materials. Based on accurate dimensional analysis, a mechanism that influences clay is proposed: the intercalation of the side chains of superplasticizer molecules in the interlayer space of the clay [17]. Moreover, the structure of Polycarboxylic contains polar groups, such as -COO-, -OH, etc., which were adsorbed on the high-energy active sites of the fine lines to reduce the free energy. In addition, these polar groups could also infiltrate into the fine grain gaps, result in weakened electrostatic effect from powders, restraint in polymerization knots and over-grinding, and improved performance of cement particle grading [18].

In this paper, polycarboxylic acids with three polar groups including carboxyl, hydroxyl, and sulfonic acid structures were synthesized by the additional polymerization of unsaturated olefin derivatives containing polar functional groups. Moreover, the influence of PC and compounds with TEA and TIPA on the cement grinding, hydration process, microstructure, and strength development were discussed.

## 2. Experiment and Characterization

### 2.1. Raw Materials

Commercially available acrylic acid (analytically pure), Sodium Allyl Sulfonate (analytically pure), and Isopentenol polyether (TPEG) were from Zhejiang, China. Deionized water, Ammonium persulphate (analytically pure), sodium hydroxide (analytically pure), triethanolamine (analytically pure), and triisopropanolamine (analytically pure) were used as raw materials. The cement clinker was from Hunan, China. The natural dihydrate gypsum (industrial grade) and Portland cement, with a strength of 42.5 Mpa (PO.42.5), were from Hunan, China. The reference cement had a strength of 42.5 Mpa.

### 2.2. Synthesis Process

The polycarboxylate grinding aids are regarded as water-soluble copolymers. They are composed of carboxyl groups and long side chains of polyoxyethylene ether. Hydrogen peroxide and vitamins are used as initiators. In each process, the polymerization was carried out in a reactor with a stirrer and a reflex condenser. First of all, 2.16 g (0.03 mol) of acrylic acid, 0.57 g (0.004 mol) of sodium allyl sulfonate, and 24 g (0.01 mol) of TPEG were dissolved in 30 g of deionized water and heated to 65 °C. Secondly, 0.16 g of ammonium persulfate aqueous solution was slowly added dropwise under nitrogen protection. The reaction was stirred for 3 h and held for 1 h. After cooling to room temperature, the pH of the solution was adjusted from 4 to 5 using a 1 mol/L aqueous solution of sodium hydroxide. A pale yellow transparent polycarboxylic acid was obtained with a solid content of 39.8%. The structure of the PC is shown in Figure 1. The polycarboxylate grinding aid (PC) was synthesized by the above method. The grinding aids were compounded with 20% TEA (GA1) and 20% TIPA (GA2), respectively, to study the effect of the above three grinding aids on the cement performance.

### 2.3. Characterization

The molecular structure of the polycarboxylate grinding aid was confirmed by FT-IR (Nicolet iS10, Thermo Scientific, Waltham, MA, USA, the infrared spectral scanning frequency range is 500~4000 cm^−1^). The PC was dried in a vacuum drying oven then mixed with KBr and pressed into disks. The particle size of cement, which was prepared with 95% clinker (which comes from Hunan, Southern of China) and 5% natural dihydrate gypsum ball milling for 25 min, was tested by laser particle sizer (Microtrac S3500, York, PA, USA). The heat release rate of cement for 48 h was detected by TAM Air Microcalorimeter. Due to the water reduction of polycarboxylic acid, the cement paste fluidity was tested. The amount of polycarboxylic acid is 0.12% to 0.24%. We recorded the flow rate at 0, 30, and 60 min, respectively. Cement compressive strength was tested by the TYA-300B microcomputer controlled constant load tester to test the compressive strength for 3 days and 28 days. Cement hydration products were characterized by X-ray diffraction (XRD, Rigaku D/max 2550, Cu Kα radiation; the working parameters are: tube pressure 40 kV, pipe flow 300 mA, step length 40 s, step width 0.02°) and their microstructure was analyzed by SEM (A Nova Nano SEM230 scanning electron microscope from FEI Company, Hillsboro, OR, U.S.A was used.).

## 3. Results and Discussion

### 3.1. Characterization of PC

Figure 2 shows the FT-IR spectra of the polycarboxylate grinding aid. The bands occurring in the FT-IR spectra of the PC can be characterized as follows: the distinct broad band at 3700~3300 cm^−1^ attributes to O-H, and the O-H stretching vibration is at 3442 cm^−1^. The O-H in plain and out-plain bending vibrations are observed at 1467 cm^−1^ and 963 cm^−1^. The strong peak observed at 2800~3000 cm^−1^ is assigned to CH_2_ stretching vibration. The out-plain bending vibration of CH_2_ is presented at about 1242 cm^−1^. The absorption observed at 1730 cm^−1^ corresponds to the C=O stretching vibration. The band at 1577 cm^−1^ is due to the presence of COO antisymmetric stretching vibrations. A strong peak observed at around 1115 cm^−1^ is assigned to C-O-C stretching vibration. The C-H stretching vibration appears at 1343 cm^−1^. The C-OH stretching vibration is normally observed at about 1280 cm^−1^. The C-S bond bending vibration is 529 cm^−1^, the characteristic peak of the sulphonic group. It indicates that functional groups, such as carboxyl group, sulfonic acid group, and polyoxyethylene ether group, have been successfully introduced into the polycarboxylic acid through the copolymerization reaction, corresponding well to the designed molecular structure as expected.

### 3.2. Particle Size Distribution

Table 1 shows the particle size distribution of ground cement without and with grinding aids at the dosage of 0.03% under the same grinding conditions. The results show that a small amount of grinding aids could improve the grindability of the cement and mitigate the particle agglomeration phenomena during the grinding process. After the introduction of the single-component PC, the amount of particles with a size ranging from 3 to 32 μm in cement could be increased by 5%, while the average particle size decreases by 3.65 μm. However, after adding GA1 and GA2, the amount of particles of 3~32 μm in cement increases by 6.31% and 9.74%, and the average particle size decreases by 4 μm and 4.7 μm, respectively. In a word, the cement particles were well distributed as the grinding aids were added. According to the distribution of particles of 3~32 μm, the compound aids showed better grinding performances than that of single PC. The composite of TIPA and PC significantly increases the particle content in the range of 3~32 μm, which is conducive to the development of cement strength.

### 3.3. Mechanism Analysis

TIPA has a unique three-dimensional structure and high polarity, so it can play a good dispersion role in the cement grinding process, weaken the agglomeration of powder, and have a better grinding effect [19]. By means of molecular simulations, O. Altun et al. [20] showed that grinding aids, such as triisopropanolamine (TIPA), triethanolamine (TEA), *N*-methyl-diisopropanolamine (MDIPA), and glycerin, orientate to a polar surface with the polar parts of the molecule, while the non-polar remainder shields the particle. They found that adsorption onto cement surfaces happens by coordination of Ca^2+^ ions and hydrogen bonds as well as other polar interactions. When the particles are very fine, the opposite charge, as mentioned above, will be agglomerated by ionic bonding. As a result, grinding efficiency will be reduced. As for the grinding aids, they could be absorbed on the surface of cement particles, neutralize the charge-screening attraction, prevent agglomeration of particles, and enhance flowability of powders, leading to an improved grinding performance. Recently, it was also shown that polymer-based grinding aids, such as polycarboxylate ethers, are very effective regarding the flowability enhancement, especially of cement powder [21]. Moreover, Haijing Yang. et al. [22] et al. considered the molecular structure of grinding aids and concluded that the chemical structure plays a decisive role in the grinding aid, which includes the type and number of functional groups and the chain length of the non-polar part within the molecule. Xu et al. [23] concluded that the adsorption behavior of clay to β-cyclodextrin polycarboxylic acid superplasticizer belongs to Ca^2+^-induced surface electrostatic adsorption, and β-cyclodextrin can prevent other polycarboxylic acid molecules from approaching concrete particles, and enhance the β-cyclodextrin. Polycarboxylic acid superplasticizer inhibits the adverse effect of concrete on its dispersion performance. In the work, the grinding mechanism is proposed as shown in Figure 3 based on the above-mentioned theories. The oxygen atom and sulphur atom in the polar group of polycarboxylic acid with strong electronegativity could attract electron pairs, and result in improved positive electricity of the hydrogen atom and sodium atom. As a result, the hydrogen bonds or ionic bonds could be formed between the hydrogen atom, sodium atom, and O^2−^ ions from the surface of cement particles. Both TEA and TIPA could form hydrogen bonds with polar groups of polycarboxylic acid and O^2−^ ions, building a stable three-dimensional network membrane, effectively preventing powder agglomeration, and improving the interparticle fluidity. In TIPA, the methyl, which is electron-donating, group in TIPA, and the oxygen atom in the hydroxyl group is more negative, and the hydrogen bond energy is greater than the TEA. This could be responsible for the improved grinding performance of cement with the presence of GA2 as grinding aid.

### 3.4. Cement Paste Fluidity

In general, polycarboxylate is widely used in cement and concrete as a water reducer. The influence of different amounts of the polycarboxylate grinding aid on the fluidity of cement paste is shown in Figure 4. Both the initial fluidity and the fluidity loss were significantly improved when the PC content increased. When the dosage of PC was 0.08%, the best water reduction was reached in cement. The initial fluidity reached 290 mm, and the fluidity at 1 h and 1.5 h was 265 mm and 260 mm, respectively, corresponding to high efficiency in water reduction. Moreover, the mechanism related to water reduction is similar to the mechanism of grinding aids, as mentioned above. The PC backbone was adsorbed on the surface of the cement particles, and hydrogen bonds were formed between the polar branches and the hydrogen atoms in the water, forming a water film on the surface of the particles, relieving agglomeration of particles, and finally, improving the fluidity of particles.

### 3.5. Cement Strength Development

Figure 5 shows the effect of grinding aids on the compressive strength of cement when the dosage of grinding aids was 0.03%. With the addition of GA1 and GA2, the compressive strength of the reference cement paste is improved significantly for 3 days and 28 days. Due to the retardation of PC, the strength of cement decreased slightly over 3 days. However, the addition of GA1 and GA2 resulted in an increase both in the early strength of cement by 6.8 MPa and 6.1 MPa, respectively, and in the later strength of cement by 5.9 MPa and 7.7 MPa. In a word, GA1 and GA2, as grinding aids, are more favorable to the evolution of cement strength. The significantly increased amount of particles of 3~32 μm is one of the main reasons for the enhanced strength [24]. 

### 3.6. Thermal Analysis during Cement Hydration and XRD Analysis

The polycarboxylate grinding aid, GA1, and GA2 were mixed into the Portland cement, respectively, in order to study the hydration exothermic rate curve, as shown in Figure 6. The third hydration peaks are observed in the cements with grinding aids. These peaks are ascribed to the exothermicity of the conversion reaction from AFt to the AFm. Grinding aids accelerate the hydration rate of C_3_A and C_4_AF by chelating with Ca^2+^ and Fe^2+^ and accelerating SO_4_^2−^ depletion. Moreover, the exothermic rate of hydration of GA1 and GA2, especially GA2, are higher than that of PC, which could significantly increase the early strength of cement.

Figure 7 shows the XRD patterns of cements after hydration for 3 and 28 days. Obviously, the diffraction patterns of cements after 3 and 28 days with the grinding aids are similar to the cement without aids, and no new phase appears in the cements with aids. As the continuous hydration reaction proceeds, the diffraction patterns of C_3_S and C_2_S in cements were weakened from day 3 to day 28, indicating decreased content of C_3_S and C_2_S, but increased contents of C-S-H and CH. Moreover, the hydration behavior of cement with grinding aids was consistent with cement without grinding aids. However, the aids could change the hydration and hydration product structure.

### 3.7. SEM Observation

Figure 8 shows the SEM morphologies of cement with or without grinding aids after 3 and 28 days. As shown in Figure 8a,b, a porous honeycomb structure composed of thin layer of CH and a large amount of curled flakes of C-S-H were presented in the pure cement and cement with PC as aids. However, compared to the cement without aids or with PC, the needle-like ettringites as reinforcements are observed in the cements with GA1 and GA2 as grinding aids after 3 days, as shown in Figure 8c,d. Note that the needle-like ettringites, as shown in Figure 8d, are significantly increased compared to that in Figure 8c, probably leading to effectively improved early strength of the cement. As the hydration extended to 28 h, many interfaces and cracks, as shown in Figure 8e, could be detected in the cement without aids. Moreover, no columnar crystals in cements are observed. However, the morphology of cements with aids is significantly different from the cements after 3 h. The cements with aids after 28 h are pretty dense, and the needle-like or hexagonal columnar crystals with very good crystallinity are encapsulated by C-S-H gels, as shown in Figure 8f–h. This effect can be explained by the SEM result that showed there were fewer fine particles absorbed on cement; thus, increasing the grinding efficiency, while PC was used as a cement grinding aid compared to a blank sample. Furthermore, the employment of PC during the grinding process increased the fluidity of cement paste significantly, while doing no obvious harm to other properties of the cement [25]. Based on the SEM morphology, the addition of grinding aids can improve the microstructure of cement hydration products and improve the strength of cement.

## 4. Conclusions

In this paper, the polycarboxylic acid with polycarboxylate grinding aid was prepared using acrylic acid, sodium allyl sulfonate, and isoprenol polyoxyethylene ether (TPEG) as raw materials, and ammonium persulfate as initiator in an N_2_ atmosphere. By studying the effects of PC and its compound triethanolamine (TEA) and triisopropanolamine (TIPA) on cement particle size, strength, hydration process, hydration products, and cement hardening slurry structure, the grinding aid mechanism of the ternary polar polycarboxylic acid grinding aid was explored. The results of the study indicate that:

The polycarboxylate grinding aid has both good grinding aid and efficient water reducing properties, delaying the accelerated period of cement hydration and generating secondary calcium alumina, which has little effect on cement strength and structure of hydration products.

The average particle diameter of cement was reduced by 3.65 μm when 0.03 wt% of PC was added as grinding aid. Moreover, high initial fluidity of the cement paste, 290 mm, could be reached when 0.08 wt% of PC was added. The fluidity loss of cement paste after 60 min and 90 min was 265 mm and 260 mm, respectively. After PC was compounded with TEA and TIPA, 4.07 μm and 4.7 μm of the average particle size of the cement can be reduced, respectively, and 5.9 MPa and 7.7 Mpa in” PO.42.5” cement in 28-day strength can be increased, respectively.

Through the analysis of the cement hydration exothermic rate, hardened slurry physical phase, and microstructure, the PC can change the cement hydration process, but has no effect on the product type, and can improve the hydration product morphology and structure.

The polycarboxylate grinding aid compounded with TEA and TIPA can improve cement grinding aid performance, significantly improve cement strength and increase the exothermic rate of hydration. Moreover, it leads to a secondary hydration acceleration period, which improves which improves the cement-hardening paste microstructure and contributes to the enhancement of cement.

## Figures and Tables

**Figure 1 polymers-14-03905-f001:**
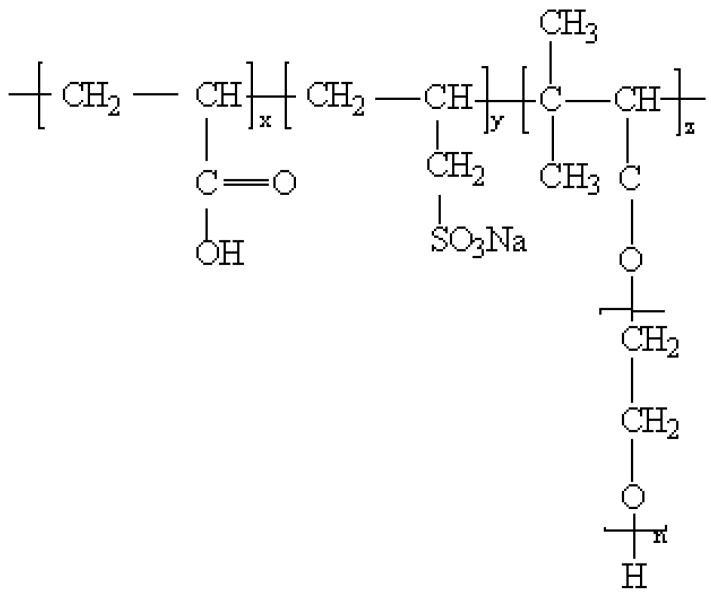
Polycarboxylic acid molecular structure.

**Figure 2 polymers-14-03905-f002:**
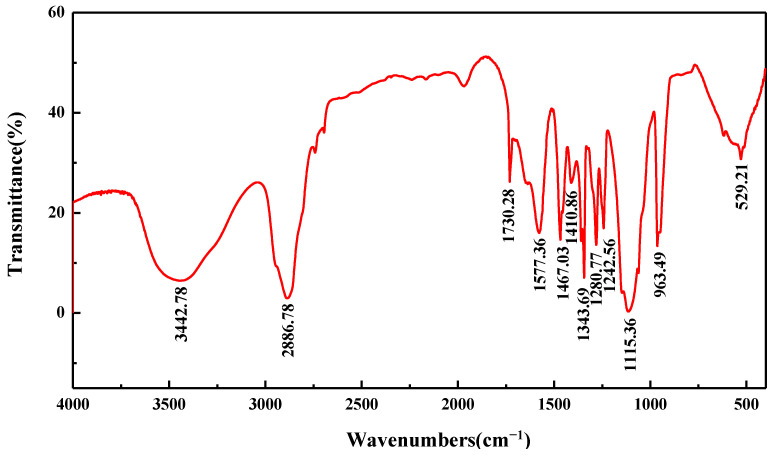
FT-IR spectra of PC.

**Figure 3 polymers-14-03905-f003:**
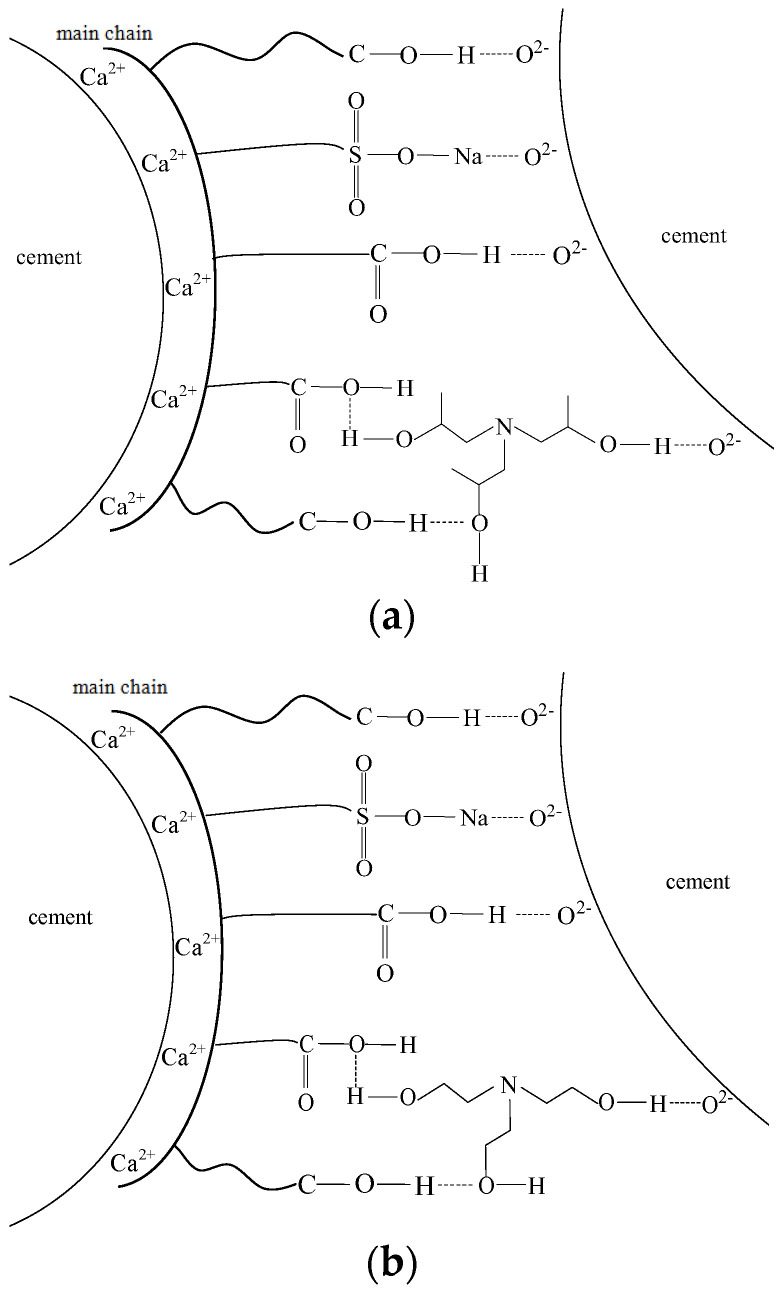
Diagram for absorption and dispersion of grinding aids on cement surface. (**a**) GA1; (**b**) GA2.

**Figure 4 polymers-14-03905-f004:**
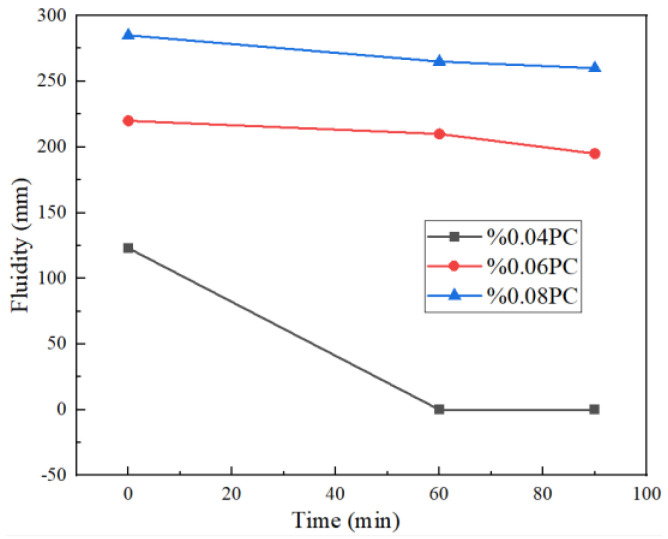
Effect of different PC content on the fluidity of cement paste.

**Figure 5 polymers-14-03905-f005:**
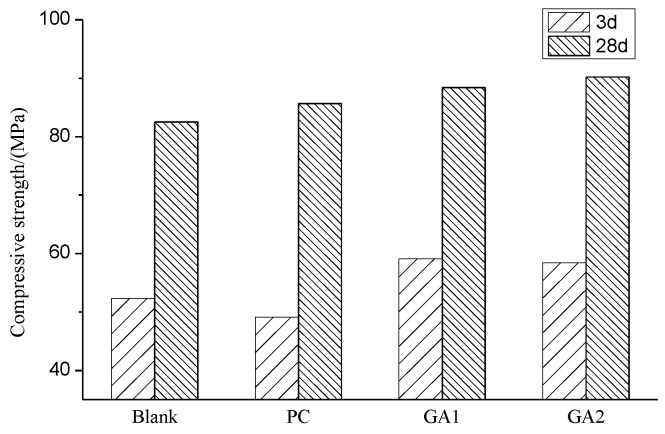
Compressive strength of cement paste for 3 days and 28 days.

**Figure 6 polymers-14-03905-f006:**
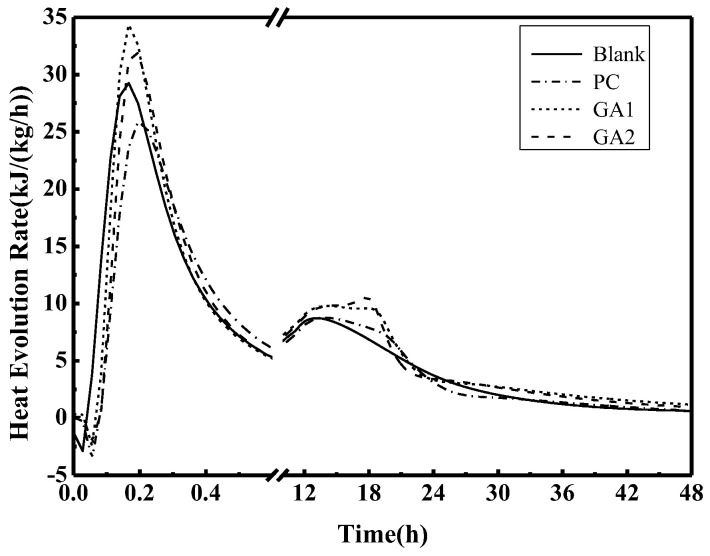
Exothermic curves of cement hydration in the first 48 h for cement using grinding aids.

**Figure 7 polymers-14-03905-f007:**
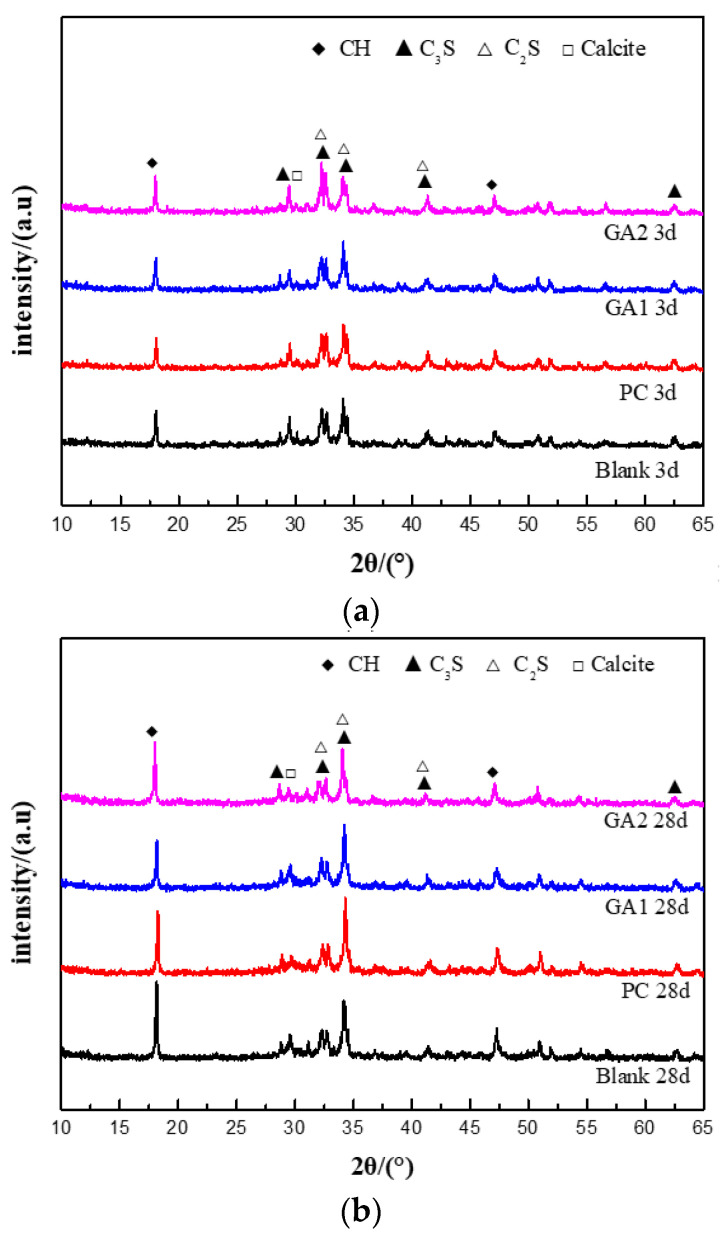
XRD patterns of cement pastes with grinding aids. (**a**) Cements after hydration for 3 d; (**b**) cements after hydration for 28 d.

**Figure 8 polymers-14-03905-f008:**
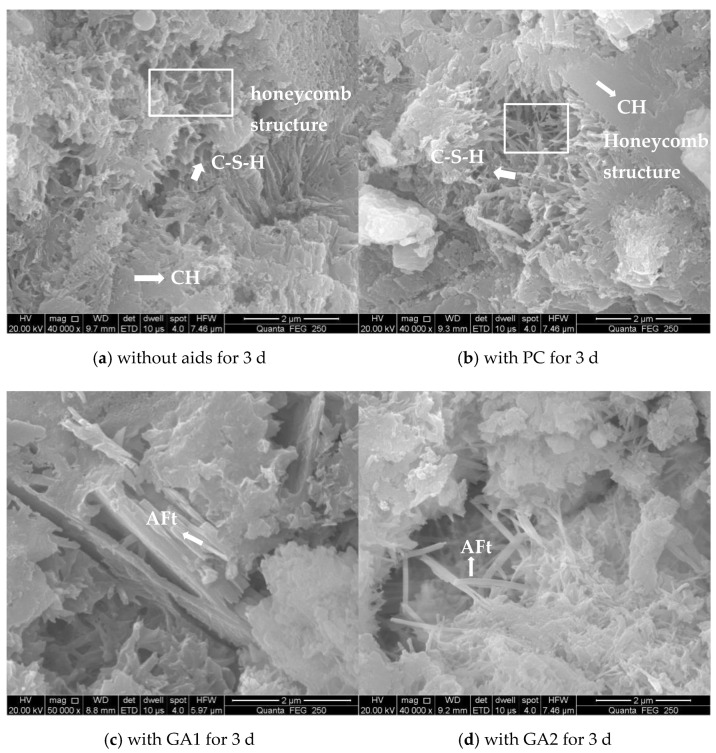
SEM morphology of cement pates with grinding aids.

**Table 1 polymers-14-03905-t001:** Particle size distribution of the composite cement grinding aids.

Grinding Aid	Particle Size Distribution (wt%)				Average Particle Size (μm)
	0–3 μm	3–32 μm	<45 μm	>65 μm	
Blank	2.70	59.57	77.07	11.39	33.28
PC	2.43	64.58	80.92	9.25	29.63
GA1	2.75	65.88	83.14	7.17	29.21
GA2	3.25	69.31	85.11	8.96	28.58

## Data Availability

The data that support the findings of this study are available from the corresponding author upon reasonable request.

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
