# Peer review of "Effect of Polycarboxylic Grinding Aid on Cement Chemistry and Properties"

_polymers, 2022, doi:10.3390/polym14183905_

Round 1
Reviewer 1 Report
The paper "Effect of polycarboxylic acid with three polar groups of the composite grinding aid on cement chemistry and properties" is very well structured and suitable for publication, I suggest its approval after minor corrections:
(a) The title is a bit confusing, the authors may review these questions;
(b) The methodological description is very insufficient, the authors must complement it with other interesting and relevant complementary information;
(c) There are parts in Chinese (note the axis of figure 2);
(d) Add error bars to the strength results;
(e) The conclusion is not entirely adequate, it should be improved;
(f) The list of references, and theoretical reference are limited, observe and consider other works of microstructural characterization, such as: 10.3390/ma14154304; 10.1016/j.cscm.2022.e00920; 10.1016/j.cscm.2022.e01155.
Author Response
Comment1: The title is a bit confusing, the authors may review these questions;
Response1:The title has been simplified and changed from “Effect of polycarboxylic acid with three polar groups of the composite grinding aid on cement chemistry and properties” to “Effect of polycarboxylic grinding aid on cement chemistry and properties”
Comment2: The methodological description is very insufficient, the authors must complement it with other interesting and relevant complementary information;
Response2:In Chapter 2.3, the performance test methods adopted are described in more detail.
Comment3:There are parts in Chinese (note the axis of figure 2);
Response3:Sorry, there was an upload error in the previous version, and the figure 2 has been modified.
Comment4: Add error bars to the strength results;
Response4:The test of cement strength in this article refers to GBT/17671-199 Method of Testing Cements—Determination of Strength in China.Three specimens were made for each group of test. we discarded the wrong test data when the test results had a large error. Moreover, the specimen was reworked and the data was remeasured.
Comment5:The conclusion is not entirely adequate, it should be improved;
Response5:I summarized the research results of the article again, and changed the original conclusion to:
In this paper ,the polycarboxylic acid with three polar groups of grinding aid (PC) was prepared using acrylic acid, sodium allyl sulfonate and isoprenol polyoxyethylene ether (TPEG) as raw materials, and ammonium persulfate as initiator in N2 atmosphere.By studying the effects of PC and its compound triethanolamine (TEA) and triisopropanolamine (TIPA) on cement particle size, strength, hydration process, hydration products and cement hardening slurry structure, the grinding aid mechanism of ternary polar polycarboxylic acid grinding aid was explored. The results of the study indicate that:
- The PChas both good grinding aid and efficient water reducing properties, delaying the accelerated period of cement hydration and generating secondary calcium alumina, which has little effect on cement strength and structure of hydration products.
- The average particle diameter of cement was reduced by 3.65 μm when 0.03 wt% of PC was added as grinding aids. Moreover, high initial fluidity of the cement paste, 290 mm could be reached while 0.08 wt% of PC was added. The fluidity loss of cement paste after 60 min and 90 min was 265 mm and 260 mm, respectively. After PC was compounded with TEA and TIPA, 4.07 μm and 4.7 μm in the average particle size of the cement can be reduced respectively and 5.9MPa and 7.7Mpa in PO.42.5 cement 28-day strength can be increased respectively.
- Through the analysis of cement hydration exothermic rate, hardened slurry physical phase and microstructure, the PC can change the cement hydration process, but has no effect on the product type, and can improve the hydration product morphology and structure
- ThePC compounding with TEA and TIPA can improve cement grinding aid performance, significantly improve cement strength, increase the exothermic rate of hydration and appear secondary hydration acceleration period, improve the microstructure of cement hardening slurry, and help increase cement strength.
Comment6:The list of references, and theoretical reference are limited, observe and consider other works of microstructural characterization, such as: 10.3390/ma14154304; 10.1016/j.cscm.2022.e00920; 10.1016/j.cscm.2022.e01155.
Response6:In this paper, some references had been added. the latest research results weree cited, which makes the article more grounded.
Reviewer 2 Report
Comments:
1. Abstract is good but still need to improve further. Please include novelty at the end of abstract
2. Please highlight main problem statement in Abstract
3. Abstract -‘.…as initiator in N2 atmosphere.’ Please mention about N2 clearly.
4. Line 41-42 ‘Investigations revealed that compounds containing more than two groups of polar present better grinding performance than those made from single polar.’- Please add related reference
5. Line 49-50 ‘Therefore, to develop grinding aids with moderate cost and good stability is currently of significance’- This statement not much related to this study. Please revise
6. Line 62-68: Please add some more related reference
7. In introduction need to highlight main problem statement clearly.
8. ‘..P.O 42.5 Portland cement, Hunan, China; P.O 42.5..’-What are the meaning of PO? Please clarify
9. ‘PC is regarded…’ - Please avoid to start sentences with Abbreviation/Number/Symbol (Check for whole article)
10. Line 102: Figure 1. Polycarboxylic acid molecular structure.- Any reference?
11. Figure 2: Please change Chinese font to English font.
12. Please add labelling to Figure 2 mention clearly which spectra related to new functional group
13. Line 145-146 ‘TIPA and PC significantly increases the particle content in the range of 3~32 μm 145 which is conducive to development of cement strength’ This statement need more discussion and justification. Please discuss how this happen?
14. Line 148:Possible mechanism - Term posibble not suitable, suggest to change
15. Line 149-161: This paragraph look refer to much with previous reference. Please highlight the finding or idea from this study first. Strongly suggest to REVISE and reduce citation from previous - this show lack of novelty.
16. Line 188-191: ‘The PC backbone was adsorbed on the……the fluidity of particles’ This statement need to justify more. How this happen. Need more critical discussion.
17. Figure 5 - Please add error bar
18. Line 245-247 ‘Based on the SEM morphology,……finding aids could modify the microstructure of hydration 246 products for cement and results in strength enhancement’ - Need more critical discussion and clarification.
19. Figure 8 - Please add labelling related to discussion.
20. Some references are more then 15 years - please use up to date reference at least 5 years back and add some more missing references
21. Conclusion need to mapping with problem statement in Introduction. Please REVISE
Author Response
Comment1: Abstract is good but still need to improve further. Please include novelty at the end of abstract
Response1:In this paper,I improved the abstract, and added novelty at the end of abstract to show the innovative content of this paper better.
Comment2: Please highlight main problem statement in Abstract
Response2:In this paper,I highlighted the main problem statement in Abstract” In view of the disadvantages of polycarboxylic acid grinding aids, such as poor reinforcement effect and cumbersome synthesis process, a new type of polycarboxylic acid grinding aid was prepared to meet the requirements of multifunctional admixture for cement concrete.”
Comment3: Abstract -‘.…as initiator in N2 atmosphere.’ Please mention about N2 clearly.
Response3:N2 is nitrogen, I have made corresponding adjustments in the text.
Comment4: Line 41-42 ‘Investigations revealed that compounds containing more than two groups of polar present better grinding performance than those made from single polar.’- Please add related reference
Response4:The two sentences, ‘Investigations revealed that compounds containing more than two groups of polar present better grinding performance than those made from single polar.’ and ‘Moreover, grinding performance is also better with the increased polar groups and the larger structure of these non-polar groups’, are contextual and have the same reference [9].
Comment5: Line 49-50 ‘Therefore, to develop grinding aids with moderate cost and good stability is currently of significance’- This statement not much related to this study. Please revise
Response5:The content of the original text is not correctly stated. I have adjusted the original text, I change ‘Additionally, TEA and TIPA, In general, were used together with inorganic salts as combinations. Note that a large quantity of inorganic salts in cement can obviously perform negative influence on the durability and safety of concrete ’ to ‘But it increased the price of the product. Additionally, TEA and TIPA, In general, were used together with inorganic salts as combinations, and a large quantity of inorganic salts in cement can obviously perform negative influence on the durability and soundness of concrete’. Beacuse a large quantity of inorganic salts in cement can obviously perform negative influence on the soundness of concrete and the combination of TEA and TIPA will increase the price of the product, so to develop grinding aids with moderate cost and good stability is currently of significance.
Comment6: Line 62-68: Please add some more related reference
Response6:I have added some related reference in this part of paper.
Comment7: In introduction need to highlight main problem statement clearly.
Response7: The content of the original introduction is not clearly, I revised the introduction to make the content more logical. I hope my revision will make the content clearer
Comment8: ‘..P.O 42.5 Portland cement, Hunan, China; P.O 42.5..’-What are the meaning of PO? Please clarify
Response8: P.O is the code name of ordinary silica cement in China. I have changed ‘P.O 42.5 Portland cement’ and ‘P.O 42.5 Reference cement’ into ‘ Portland cement with strength of 42.5Mpa’ and ‘Reference cement with strength of 42.5Mpa’ in the text.
Comment9: ‘PC is regarded…’ - Please avoid to start sentences with Abbreviation/Number/Symbol (Check for whole article)
Response9: Adjustments have been made in the text, ’PC’ has been changed to ‘polycarboxylate grinding aid’.
Comment10: Line 102: Figure 1. Polycarboxylic acid molecular structure.- Any reference?
Response10: Polycarboxylic acid molecular structure is derived from the synthesis of acrylic acid, sodium allyl sulfonate and so on. This is an expected result. As the result, there are no reference.
Comment11: Figure 2: Please change Chinese font to English font.
Response11: Sorry, there was an upload error in the previous version, and the figure 2 has been modified.
Comment12: Please add labelling to Figure 2 mention clearly which spectra related to new functional group
Response12: In Figure 2, some numbers are marked, and relationship between these numbers and new functional group was analysed in paper.
Comment13: Line 145-146 ‘TIPA and PC significantly increases the particle content in the range of 3~32 μm 145 which is conducive to development of cement strength’ This statement need more discussion and justification. Please discuss how this happen?
Response13: The analysis results of this statement is similar to journal article [23], and this article has been cited in the paper.
Comment14: Line 148:Possible mechanism - Term posibble not suitable, suggest to change
Response14: In the paper ‘Possible mechanism’ has changed to ‘Mechanism analysis’.
Comment15: Line 149-161: This paragraph look refer to much with previous reference. Please highlight the finding or idea from this study first. Strongly suggest to REVISE and reduce citation from previous - this show lack of novelty.
Response15: In this part, I made corresponding modifications, added new references, and adjusted the content expression
Comment16: Line 188-191: ‘The PC backbone was adsorbed on the……the fluidity of particles’ This statement need to justify more. How this happen. Need more critical discussion.
Response16: I deleted this part because the purpose of this analysis is to explain the influence of PC content on the fluidity of cement paste.
Comment17: Figure 5 - Please add error bar
Response17: The test of cement strength in this article refers to GBT/17671-199 Method of Testing Cements—Determination of Strength in China.Three specimens were made for each group of test. we discarded the wrong test data when the test results had a large error. Moreover, the specimen was reworked and the data was remeasured.
Comment18: Line 245-247 ‘Based on the SEM morphology,……finding aids could modify the microstructure of hydration 246 products for cement and results in strength enhancement’ - Need more critical discussion and clarification.
Response18:In the article, I have added more analysis, discussion and references
Comment19: Figure 8 - Please add labelling related to discussion.
Response19: Corresponding changes have been made
Comment20: Some references are more then 15 years - please use up to date reference at least 5 years back and add some more missing references
Response20: I have updated some older references
Comment21: Conclusion need to mapping with problem statement in Introduction. Please REVISE
Response21: I summarized the research results of the article again, and changed the original conclusion to:
In this paper ,the polycarboxylic acid with three polar groups of grinding aid (PC) was prepared using acrylic acid, sodium allyl sulfonate and isoprenol polyoxyethylene ether (TPEG) as raw materials, and ammonium persulfate as initiator in N2 atmosphere.By studying the effects of PC and its compound triethanolamine (TEA) and triisopropanolamine (TIPA) on cement particle size, strength, hydration process, hydration products and cement hardening slurry structure, the grinding aid mechanism of ternary polar polycarboxylic acid grinding aid was explored. The results of the study indicate that:
- The PChas both good grinding aid and efficient water reducing properties, delaying the accelerated period of cement hydration and generating secondary calcium alumina, which has little effect on cement strength and structure of hydration products.
- The average particle diameter of cement was reduced by 3.65 μm when 0.03 wt% of PC was added as grinding aids. Moreover, high initial fluidity of the cement paste, 290 mm could be reached while 0.08 wt% of PC was added. The fluidity loss of cement paste after 60 min and 90 min was 265 mm and 260 mm, respectively. After PC was compounded with TEA and TIPA, 4.07 μm and 4.7 μm in the average particle size of the cement can be reduced respectively and 5.9MPa and 7.7Mpa in PO.42.5 cement 28-day strength can be increased respectively.
- Through the analysis of cement hydration exothermic rate, hardened slurry physical phase and microstructure, the PC can change the cement hydration process, but has no effect on the product type, and can improve the hydration product morphology and structure
- ThePC compounding with TEA and TIPA can improve cement grinding aid performance, significantly improve cement strength, increase the exothermic rate of hydration and appear secondary hydration acceleration period, improve the microstructure of cement hardening slurry, and help increase cement strength.

Round 2
Reviewer 2 Report
Author has made correction based on comments. Small changes still need to made:
1. Figure 2 - Please check unclear text under wavenumbers title
2. Please add labelling based on the discussion for Figure 8.
3. Please replace references other tan 15 years with updated references
Author Response
Comment1: Figure 2 - Please check unclear text under wavenumbers title
Response1: has been modified.
Comment2:Please add labelling based on the discussion for Figure 8.
Response2:The labelling has been added on Figure 8.
Comment3:Please replace references other tan 15 years with updated references
Response3:The references have been updated.